# Primary and Secondary Tumors of the Parotid Gland: Clinical Features and Prognosis

**DOI:** 10.3390/cancers15041293

**Published:** 2023-02-17

**Authors:** Giancarlo Pecorari, Claudia Pizzo, Marco Briguglio, Ester Cravero, Giuseppe Riva

**Affiliations:** Division of Otorhinolaryngology, Department of Surgical Sciences, University of Turin, 10126 Turin, Italy

**Keywords:** salivary gland, parotid gland, skin cancer, salivary gland tumors, parotid tumors, intraparotid metastasis

## Abstract

**Simple Summary:**

Malignant salivary gland tumors are rare, accounting for 0.05–2 per 100,000 inhabitants worldwide. The parotid gland represents the most frequent cancer site, and both primary and secondary malignant tumors can affect it. Primary cancer includes many histological types, while intraparotid metastases have usually a cutaneous origin. Because of the rarity and heterogeneity of parotid tumors, the prognosis is difficult to assess. The literature focused on specific primary tumors or secondary cancer. The aim of this retrospective observational study was to evaluate the clinical features and prognosis of malignant epithelial tumors of the parotid gland. In particular, a comparison between primary and secondary cancer and survival analyses were performed. Moreover, complications were recorded.

**Abstract:**

Primary and secondary malignant tumors can affect the parotid gland. The aim of this retrospective study was to evaluate the clinical features and prognosis of malignant epithelial tumors of the parotid gland. In particular, a comparison between primary and secondary cancer and survival analyses were performed. Eighteen patients with primary cancer and fifteen with intraparotid metastasis from cutaneous squamous cell carcinoma were included. A chart review was performed to collect clinical data (age, sex, smoking, alcohol consumption, tumor stage, type of surgical procedure, complications, recurrence and death). The majority of primary tumors were early (T1-2 N0, 83%) with mucoepidermoid carcinoma being the most common (33%). Secondary tumors were mostly staged P2 (53%) and N0 (67%). Subjects with secondary tumors were older than those with primary cancer. Post-operative permanent facial palsy was observed in 5 patients (17%) with primary cancer and 9 (60%) with secondary tumors (*p* = 0.010). Two-year overall survival for primary and secondary parotid cancer was 76.58% and 43.51%, respectively (*p* = 0.048), while 2-year disease-free survival was 76.05% and 38.50%, respectively (*p* = 0.152). In conclusion, secondary cancer of the parotid gland has worse survival than primary tumors. In the future, the implementation of multimodality treatment of intraparotid metastases is necessary to improve oncologic outcomes.

## 1. Introduction

Malignant salivary gland tumors are rare, accounting for 0.05–2 per 100,000 inhabitants worldwide [1]. The parotid gland represents the most frequent cancer site. Nutrition, irradiation, and a long-standing histologically benign cancer may be risk factors [1]. Diagnosis is based on an imaging scan (ultrasound, computed tomography, and/or magnetic resonance imaging) and fine needle aspiration cytology or core needle biopsy. Positron emission tomography may be useful if an intraparotid metastasis is suspected.

Both primary and secondary malignant tumors can affect the parotid gland [2]. Primary cancer includes many histological types, while intraparotid metastases usually have a cutaneous origin [3,4]. Because of the rarity and heterogeneity of parotid tumors, the prognosis is difficult to assess. The literature focused on specific primary tumors or secondary cancer. Only two studies compared the overall survival (OS) of primary and secondary parotid tumors [5,6]. Jering et al. reported a 5-year cancer-specific survival of 87.3% among patients with primary cancer and 54.5% among patients with metastatic cutaneous squamous cell carcinoma [5]. Meyer et al. found a higher 5-year OS in subjects with primary cancer (77.2% vs. 32.6%) [6]. However, all patients with intraparotid metastases from cutaneous squamous cell carcinoma included in such a study had neck node involvement. Including subjects without neck metastases can allow us to perform a more realistic survival analysis. Indeed, prognostic factors in secondary parotid cancer include immunodepression, age, positive margins, facial nerve involvement, and neck nodes [7,8]. Moreover, complications of parotidectomy have never been compared to evaluating primary and secondary tumors.

The aim of this retrospective observational study was to evaluate the clinical features and prognosis of malignant epithelial tumors of the parotid gland. In particular, a comparison between primary and secondary cancer and survival analyses were performed. Moreover, complications were recorded.

## 2. Materials and Methods

Between January 2018 and June 2022, 33 patients with malignant epithelial tumor of the parotid gland were surgically treated at our department. Exclusion criteria were age < 18 years, benign tumors and non-epithelial tumors (e.g., sarcoma, lymphoma). A chart review was performed to collect clinical data (age, sex, smoking, alcohol consumption, tumor stage, type of surgical procedure, complications, recurrence and death). Parotid (P) stage was assessed for secondary cancer. In particular, P0 indicated no disease in the parotid, P1 metastatic node < 3 cm, P2 metastatic node between 3 and 6 cm, or multiple nodes, and P3 metastatic node > 6 cm, or disease involving the facial nerve or skull base [9]. Neck nodes were also classified based on O’Brien’s staging system: N0, no neck disease; N1, single ipsilateral neck node < 3 cm; and N2, single neck node > 3 cm or multiple nodes or contralateral nodes [9]. An R0 resection was the objective of the surgical procedure. The facial nerve was not preserved if pre- or intra-operative signs of its involvement was present with impossibility to achieve oncologic radicality with nerve sparing.

The patients were divided into two groups: primary and secondary tumors (18 and 15 patients, respectively). The study was conducted in accordance with the Declaration of Helsinki and approved by the Institutional Review Board (protocol code 0021433, date of approval 26 February 2021). Written informed consent was obtained.

All statistical analyses were carried out using the Statistical Package for Social Sciences, version 20.0 (IBM Corporation, Armonk, NY, USA), and GraphPad Prism, version 5 (GraphPad Software Inc., San Diego, CA, USA). A descriptive analysis of all data was performed, and they were reported as means, medians or percentages and standard deviations. The Kolmogorov-Smirnov test demonstrated a non-Gaussian distribution of variables, so non-parametric tests were used. The Mann-Whitney U test was used to assess differences between groups in the mean of continuous variables, while the chi-squared test was used for categorical variables. The Kaplan-Meier method was used for Overall Survival (OS) and Disease-Free Survival (DFS), and the Log-rank test was used for survival analyses. The endpoints were the length of time from diagnosis to death by any cause for OS, and from diagnosis to recurrence or death for DFS. A *p* value less than 0.05 was considered statistically significant.

## 3. Results

Eighteen patients with primary tumor of the parotid gland and fifteen with secondary cancer were included in the study. The most prevalent primary tumor was mucoepidermoid carcinoma (Figure 1). All secondary tumors were intraparotid metastases from cutaneous squamous cell carcinoma (SCC). No subject had immunosuppression.

Subjects with secondary tumors were older than those with primary cancer (mean age 80.60 ± 6.77 years and 61.17 ± 17.32 years, respectively, *p* < 0.001). No statistically significant difference was observed for other clinical features (Table 1). However, a higher percentage of former smokers was present in the secondary cancer group (*p* = 0.082). Maximum dimension of the parotid lesion was lower for primary tumors (2.12 ± 1.33 mm and 3.79 ± 1.19 mm, respectively, *p* < 0.001).

The most frequent surgical procedure was total parotidectomy in both groups (Table 2). Neck dissection was not performed in some cases because of a pre-operative cytological diagnosis of benign pathology and/or post-operative indication to adjuvant radiotherapy or follow-up by the multidisciplinary tumor board (age, comorbidities and anesthesiologic risks were taken into account). Adjuvant radiotherapy or chemoradiotherapy was administered to 12 patients (67%) with primary cancer and 6 patients (40%) with secondary cancer.

Among subjects with intraparotid metastasis, the removal of cutaneous SCC was concomitant to parotidectomy and neck dissection in 3 cases (20%). In the other cases, the mean time between the primary skin cancer removal and the parotidectomy was 13.60 ± 10.82 months (range 3–38 months). The site of the primary skin carcinoma was the temporal-frontal region (7 patients), the external ear (6 patients) and the cheek (2 patients).

Statistical analyses did not highlight significant differences concerning pathological characteristics between primary and secondary parotid tumors (Table 3).

Post-operative permanent facial palsy was observed in 5 patients (17%) with primary cancer and 9 (60%) with secondary tumors (*p* = 0.010). In particular, such sequela was observed in those subjects who had pre- or intra-operative signs of nerve involvement or impossibility to achieve oncologic radicality with nerve sparing. No permanent palsy was observed in nerve-preserving surgery. Transient facial palsy was reported in 5 patients (17%) with primary cancer and 3 (20%) with intraparotid metastases (*p* = 0.604, Figure 2). Other complications were two sialoceles in patients with primary tumors and two wound dehiscence in subjects with secondary cancer. Two cases of partial superficial necrosis of the deltopectoral flap were observed and solved with second intention of healing or skin graft.

The mean follow-up was 17.75 months for primary tumors and 18.26 months for secondary tumors. Two-year OS for primary and secondary parotid cancer was 76.58% and 43.51%, respectively (*p* = 0.048), while 2-year DFS was 76.05% and 38.50%, respectively (*p* = 0.152) (Figure 3).

## 4. Discussion

The prognosis of malignant parotid cancer mainly depends on tumor stage, histology, grading, facial nerve paralysis, extra-parotid tumor extension and cervical node involvement [1]. Surgery is the standard treatment for resectable tumors. Preservation of the facial nerve should be warranted whenever possible (pre- and intra-operative evidence that the nerve is not involved with cancer). Adjuvant radiotherapy or chemoradiotherapy is administered if adverse pathological features are present (high-grade tumors, perineural and/or vascular invasion, close or positive margins, advanced stage). For inoperable/unresectable tumors, exclusive radiotherapy and chemoradiotherapy are the treatment options [1].

Head and neck cutaneous SCC is the main responsible for intraparotid metastases [3,4]. In particular, metastasis to the intra- and peri-parotid lymph nodes represents the most common cause of parotid malignancy [10]. Parotid lymph nodes are the primary sites of drainage from the forehead, scalp, cheek and external ear toward the deep cervical nodes. The American Joint Committee of Cancer (AJCC) staging manual for skin SCC does not include a specific classification for intraparotid metastasis. Therefore, since the latter have near always an extranodal extension, the AJCC stage should be N3b in most cases. O’Brien et al. proposed a new classification for parotid involvement to achieve a cancer stage that was more related to prognosis [9]. This system classified metastatic SSC according to the number and size of the parotid (P) and cervical (N) metastases, and the facial nerve/skull base involvement [9,11].

Our study compared clinical characteristics, complications of surgery and prognosis of primary and secondary parotid tumors. In agreement with the literature [3,6], we found that patients with secondary cancer were older and most frequently male. The prevalence of males is due to the higher incidence of skin SCC in this sex [12]. Smoking habits and alcohol consumption were not significantly different between the two groups. The majority of subjects with primary tumors were staged early (T1-2 N0, 83%) with mucoepidermoid carcinoma being the most common histology (33%), in agreement with previous studies [13,14]. According to O’Brien’s classification, secondary tumors were mostly staged P2 (53%) and N0 (67%). This suggests that skin SCC usually determines intraparotid metastases between 3 and 6 cm in diameter without neck node involvement. However, our study showed that neck node metastases are less common in primary parotid cancer (17% vs. 33%).

Intraparotid metastases usually occur 1 to 3 years after surgical removal of the primary skin SCC [15,16,17]. We found that concomitant parotid metastases were present in 20% of subjects, while in the other cases, the mean time between the primary skin cancer removal and the parotidectomy was 13.60 months, with a range between 3 and 38 months.

The extent of parotidectomy for malignant tumors is still controversial [18]. Total parotidectomy is the most used procedure. The deep lobe should be removed if the tumor is located or involved in the deep lobe or if the tumor has spread to superficial intraparotid or cervical nodes and in cases of high-grade cancer [19]. The American Head and Neck Society suggests that parotidectomy is indicated for subjects with a clinically evident parotid disease and that the extent of parotidectomy must be based on pre-operative imaging and intraoperative findings. Furthermore, in P+ subjects, elective neck dissection or radiotherapy should be performed given the significant risk of occult metastases [20].

Pre-operative facial palsy was similar between the groups and affected about one-third of the patients. Total parotidectomy was the most frequent surgical procedure in both groups (about 80% of our cases). However, the sacrifice of the facial nerve was higher for intraparotid metastases because of the intra-operative finding of infiltration or the impossibility of achieving radicality with nerve sparing. Therefore, a higher percentage of post-operative permanent facial palsy was observed among patients with secondary tumors. This led to the need for facial reanimation surgery. The incidence of facial nerve sacrifice varies across studies in the literature (5–43%) and reflects different tumor sites and stages [21,22,23,24]. Our high rate of post-operative permanent facial palsy in patients with intraparotid metastases was related to the high percentage of locally advanced tumors (P stage).

Skin infiltration and removal were more frequent in SCC metastases than in primary parotid tumors (33% vs. 6%). Therefore, the need for flap reconstruction was higher in secondary tumors. A deltopectoral flap was used in our cases with good results (no complete necrosis).

Five- and 10-year disease-specific survivals for primary parotid cancer vary between 55–82% and 47–69%, respectively [19]. Such heterogeneous outcomes are related to the high number of different histology of parotid tumors. OS for parotid metastasis from skin SCC is usually poor. Indeed, 5-year OS ranged from 30 to 60%, also because of a more advanced age and tumor stage [25,26]. Jering et al. reported a 5-year cancer-specific survival of 87.3% among patients with primary parotid cancer and 54.5% among patients with metastatic cutaneous SCC [5]. Meyer et al. compared primary and secondary parotid tumors highlighting a 5-year OS of 77.2% and 32.6%, respectively. Moreover, they found better survival for N0 patients with primary cancer. Only a favorable trend for N0 was observed in secondary cancer [6]. Girardi et al. showed that patients presenting with isolated parotid metastasis during follow-up had better disease-specific survival than those with neck metastasis or both [27].

Our study showed a better OS for primary cancer, while DFS was not statistically different. Nevertheless, DFS was poor in patients with intraparotid metastasis. The lack of statistical significance is probably related to the small samples. Our results are in line with previous studies and are related to a generally higher tumor stage as a consequence of lymph node metastasis in the skin SCC group [5,6]. Immunosuppression is an independent prognostic indicator for reduced survival in subjects with intraparotid SCC metastasis [28]. In our case series, no immunocompromised patient was present. In the future, the further development of immunotherapy for cutaneous SCC, such as cemiplimab, may improve survival.

The limitations of this study were the retrospective nature and the low number of subjects. In particular, the small sample did not allow for a deeper analysis of prognostic factors for survival and recurrence risk. Further studies with larger samples will allow the stratification for potential prognostic factors (e.g., tumor and nodal staging, tumor grading, vascular and perineural invasion, margins) and a more detailed analysis of survival rates. These analyses may help clinicians with adjuvant treatments. The strength was the comprehensive comparison of clinical features, surgical complications and prognosis of primary and secondary parotid tumors. Indeed, previous studies analyzed only primary or secondary parotid tumors or rarely compared survival rates without considering all the clinical features and post-operative complications.

## 5. Conclusions

Malignant tumors of the parotid gland represent a diagnostic and therapeutic challenge. An appropriate examination looking for skin lesions is mandatory when a parotid SCC is diagnosed. Surgery is the main treatment and facial palsy is the most common complication. Secondary cancer of the parotid gland has worse survival than primary tumors. In the future, the implementation of multimodality treatments for intraparotid metastasis is necessary to improve oncologic outcomes. 

## Figures and Tables

**Figure 1 cancers-15-01293-f001:**
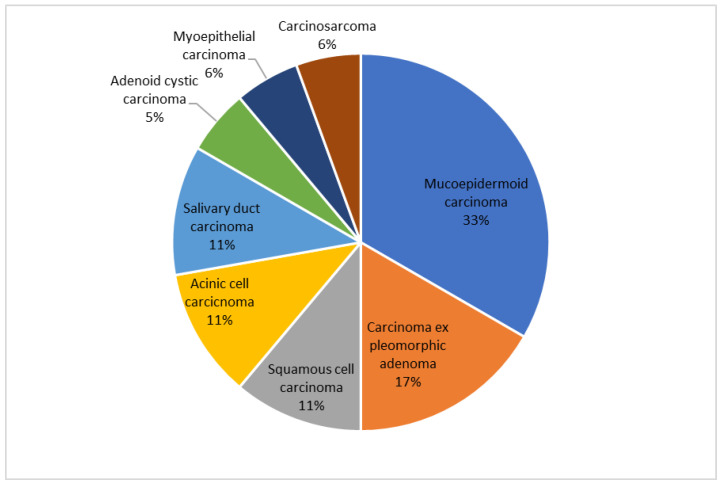
Primary tumors of the parotid gland.

**Figure 2 cancers-15-01293-f002:**
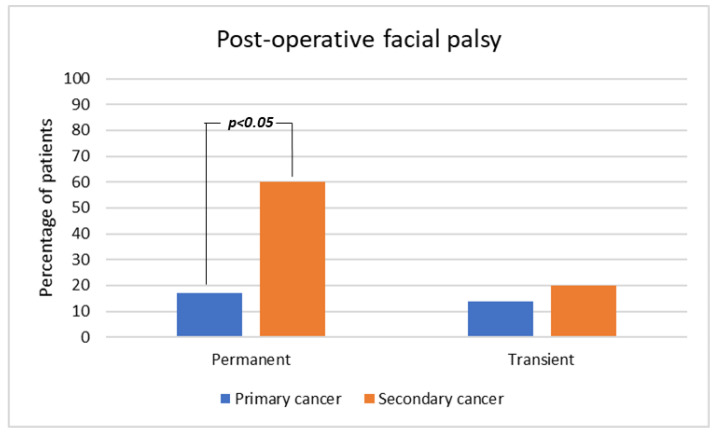
Post-operative facial palsy.

**Figure 3 cancers-15-01293-f003:**
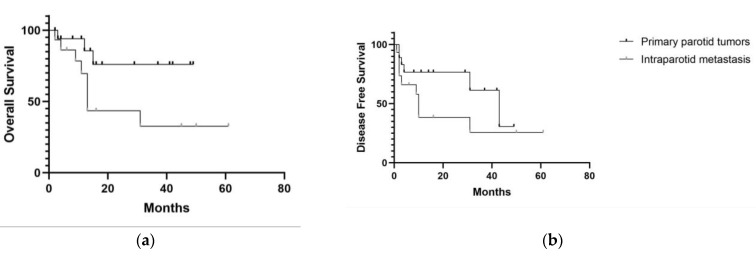
(**a**) Overall Survival for primary and secondary cancer; (**b**) Disease-Free Survival for primary and secondary cancer.

**Table 1 cancers-15-01293-t001:** Clinical characteristics (*n*, %).

Clinical Characteristics	Primary Cancer	Secondary Cancer	*p* Values
** *Sex* **			0.138
Male	10 (55.6)	12 (80.0)	
Female	8 (44.4)	3 (20.0)	
** *Smoking* **			0.082
Never	10 (55.6)	4 (26.7)	
Former	5 (27.8)	10 (66.7)	
Active	3 (16.7)	1 (6.7)	
** *Alcohol* ** ***consumption** **	3 (16.7)	3 (20.0)	0.805
** *T* **			-
0	0 (0.0)	12 (80.0)	
1	8 (44.4)	0 (0.0)	
2	5 (27.8)	1 (6.7)	
3	1 (5.6)	2 (13.3)	
4	4 (22.2)	0 (0.0)	
** *P* **			-
1	-	2 (13.3)	
2	-	8 (53.3)	
3	-	5 (33.3)	
** *N *** **			-
0	15 (83.3)	10 (66.7)	
1	0 (0.0)	0 (0.0)	
2	2 (11.1)	5 (33.3)	
3	1 (5.6)	-	
** *M* **			-
0	18 (100.0)	15 (100.0)	
1	0 (0.0)	0 (0.0)	
** *Pre-operative facial palsy* **	5 (27.8)	5 (33.3)	0.730

* Refers to current alcohol consumption (>2 drinks per day for men and >1 drink per day for women). ** N stage was based on radiologic features (clinical N stage) in 8 patients with primary cancer who did not undergo neck dissection. *p* values for the stage (T, P, N and M) were not evaluated because of the different primary sites of the tumors. T, tumor; M, metastasis; N, node; P, parotid.

**Table 2 cancers-15-01293-t002:** Surgical procedures (*n*, %).

Surgery	Primary Cancer	Secondary Cancer
Superficial parotidectomy	3 (16.7)	3 (20.0)
Total parotidectomy	14 (77.8)	7 (46.7)
Total parotidectomy with skin removal	1 (5.6)	5 (33.3)
Neck dissection	10 (55.6)	12 (80.0)
Reconstruction with deltopectoral flap	1 (5.6)	5 (33.3)

**Table 3 cancers-15-01293-t003:** Pathological characteristics (*n*, %).

Pathological Characteristics	Primary Cancer	Secondary Cancer	*p* Values
** *Grade* **			0.072
1	7 (38.9)	1 (6.7)	
2	3 (16.7)	6 (40.0)	
3	8 (44.4)	8 (53.3)	
** *Margins* **			0.887
Negative	7 (38.9)	7 (46.7)	
Close	6 (33.3)	4 (26.7)	
Positive	5 (27.8)	4 (26.7)	
** *Lymphovascular invasion* **	4 (22.2)	5 (33.3)	0.475
** *Perineural invasion* **	8 (44.4)	5 (33.3)	0.515
** *Nodal extracapsular spread* **	3 (16.7)	2 (13.3)	0.790

## Data Availability

The data presented in this study are available on request from the corresponding author.

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
