# Peer review of "Primary and Secondary Tumors of the Parotid Gland: Clinical Features and Prognosis"

_cancers, 2023, doi:10.3390/cancers15041293_

Round 1
Reviewer 1 Report
Thank you for inviting me to review the article. This compares clinical characteristics, therapy, and prognosis of primary salivary gland malignancies with metastases from squamous cell carcinoma of the skin.
On the positive side, the clear presentation of the data and the completeness of the clinical characteristics to be studied should be noted. Overall, however, the study offers little new information compared with previously published work. The authors would need to elaborate significantly more on what information has not been presented previously in the literature.
Another weakness of the study is the small number of cases, which makes statistical conclusions difficult.
The authors present a high number of permanent facial nerve palsies after surgery, although it remains open how many of these patients already had facial nerve palsies preoperatively and how many patients had a (proportional) facial resection for R0 resection on purpose. It would be desirable if the authors would describe in detail their surgical strategy for malignant tumors of the parotid gland for this purpose.
In case of paresis where nerve preserving surgery was performed, the number of permanent paresis is very high and possibly the addition of optical aids (microscope, magnifying glasses) should be considered if not performed so far.
In the current form, a rejection is recommended in my opinion.
Author Response
Thank you for inviting me to review the article. This compares clinical characteristics, therapy, and prognosis of primary salivary gland malignancies with metastases from squamous cell carcinoma of the skin.
On the positive side, the clear presentation of the data and the completeness of the clinical characteristics to be studied should be noted. Overall, however, the study offers little new information compared with previously published work. The authors would need to elaborate significantly more on what information has not been presented previously in the literature.
Thanks for your comments and suggestions. We better elaborated what information are new (strength of the study at the end of the Discussion).
Another weakness of the study is the small number of cases, which makes statistical conclusions difficult.
The small sample was described as a limitation of the study in the Discussion.
The authors present a high number of permanent facial nerve palsies after surgery, although it remains open how many of these patients already had facial nerve palsies preoperatively and how many patients had a (proportional) facial resection for R0 resection on purpose. It would be desirable if the authors would describe in detail their surgical strategy for malignant tumors of the parotid gland for this purpose.
In case of paresis where nerve preserving surgery was performed, the number of permanent paresis is very high and possibly the addition of optical aids (microscope, magnifying glasses) should be considered if not performed so far.
No permanent paresis was observed in nerve preserving surgery. All the post-operative facial palsy were related to nerve sacrifice because of its involvement by the tumor. Magnifying glasses were used. A better description of surgical startegy was added in materials and methods section.
Reviewer 2 Report
Well written paper comparing primary and secondary tumors of parotid gland, however limited to one department and includes small number of subjects.
The introduction can be made more robust by citing more relevant literature. Please check these studies too:
Jering M, Mayer M, Thölken R, Schiele S, Müller G, Zenk J. Cancer-specific and overall survival of patients with primary and metastatic malignancies of the parotid gland - A retrospective study. J Craniomaxillofac Surg. 2022 May;50(5):456-461.
Saravakos P, Kourtidis S, Hartwein J, Preyer S. Parotid Gland Tumors: A Multicenter Analysis of 1020 Cases. Increasing Incidence of Warthin's Tumor. Indian J Otolaryngol Head Neck Surg. 2022 Oct;74(Suppl 2):2033-2040.
A more extensive discussion that compares the current analyses to other studies would be useful.
Please expand the limitations and the future steps section.
Author Response
Well written paper comparing primary and secondary tumors of parotid gland, however limited to one department and includes small number of subjects.
Thanks for your positive comments.
The introduction can be made more robust by citing more relevant literature. Please check these studies too:
Jering M, Mayer M, Thölken R, Schiele S, Müller G, Zenk J. Cancer-specific and overall survival of patients with primary and metastatic malignancies of the parotid gland - A retrospective study. J Craniomaxillofac Surg. 2022 May;50(5):456-461.
Saravakos P, Kourtidis S, Hartwein J, Preyer S. Parotid Gland Tumors: A Multicenter Analysis of 1020 Cases. Increasing Incidence of Warthin's Tumor. Indian J Otolaryngol Head Neck Surg. 2022 Oct;74(Suppl 2):2033-2040.
Introduction was made more robust with the suggested references.
A more extensive discussion that compares the current analyses to other studies would be useful.
We added a more extensive discussion comparing current analyses to other studies.
Please expand the limitations and the future steps section.
We expanded the limitations and the future steps section.
Reviewer 3 Report
The paper "Primary and secondary tumors of the parotid gland: clinical features and prognosis" highlights the problem of metastatic tumours diagnosed in the parotid glands. Especially, surgeons treating patients with skin cancer should be aware of possibility of intraparotid metastases of this type of cancer. Obtained results showed worse prognosis for secondary tumours. The major drawback of this study is relatively small group of patients.
From oncological point of view, using the term 'benign cancer' (line 39) is unacceptable. Histologically, these are low-grade cancers (low/high grade mucoepidermoid cancer).
References comprise 12 papers older than 10 years. Apart from source article on O'Brien classification, papers dealing with methods of treatment should be updated, as in oncology, major changes in the therapeutic algorithms have occurred.
Also, correction of English grammar is necessary (e.g. line 109: Neck dissection were not performed).
Author Response
The paper "Primary and secondary tumors of the parotid gland: clinical features and prognosis" highlights the problem of metastatic tumours diagnosed in the parotid glands. Especially, surgeons treating patients with skin cancer should be aware of possibility of intraparotid metastases of this type of cancer. Obtained results showed worse prognosis for secondary tumours. The major drawback of this study is relatively small group of patients.
Thenks for your suggestions. The small number of patients was described a limitation of the study in the Discussion section.
From oncological point of view, using the term 'benign cancer' (line 39) is unacceptable. Histologically, these are low-grade cancers (low/high grade mucoepidermoid cancer).
The sentence at line 39 refers to histologically benign tumors, such as pleomorphic adenoma, that may trasform into malignant one.
References comprise 12 papers older than 10 years. Apart from source article on O'Brien classification, papers dealing with methods of treatment should be updated, as in oncology, major changes in the therapeutic algorithms have occurred.
More recent references were added (n. 2 and 5).
Also, correction of English grammar is necessary (e.g. line 109: Neck dissection were not performed).
Grammatical errors were corrected.
Round 2
Reviewer 1 Report
The Authors improved the manuscript. The manuscript is acceptable in the present form. A disadvantage remains that the number of patients is small and the statistical power is limited.